

# DNA methylome signatures as epigenetic biomarkers of hexanal associated with lung toxicity

Yoon Cho[1], Mi-Kyung Song[2] and Jae-Chun Ryu[1]

[1] Korea Institute of Science and Technology, Seoul, Republic of Korea
[2] Korea Institute of Toxicology, Jeongeup, Republic of Korea

## ABSTRACT

**Background**. Numerous studies have investigated the relationship of environmental exposure, epigenetic effects, and human diseases. These linkages may contribute to the potential toxicity mechanisms of environmental chemicals. Here, we investigated the epigenetic pulmonary response of hexanal, a major indoor irritant, following inhalation exposure in F-344 rats.

**Methods**. Based on DNA methylation profiling in gene promoter regions, we identified hexanal-characterized methylated sites and target genes using an unpaired t-test with a fold-change cutoff of $\geq 3.0$ and a $p$-value $< 0.05$. We also conducted an integrated analysis of DNA methylation and mRNA expression data to identify core anti-correlated target genes of hexanal exposure. To further investigate the potential key biological processes and pathways of core DNA methylated target genes, Gene Ontology and Kyoto Encyclopedia of Genes and Genomes pathway enrichment analysis were performed.

**Results**. Thirty-six dose-dependent methylated genes and anti-correlated target genes of DNA methylation and mRNA in lung tissue of hexanal exposed F-344 rats were identified. These genes were involved in diverse biological processes such as neuroactive ligand-receptor interaction, protein kinase cascade, and intracellular signaling cascade associated with pulmonary toxicity. These results suggest that novel DNA methylation-based epigenetic biomarkers of exposure to hexanal and elucidate the potential pulmonary toxicological mechanisms of action of hexanal.

## INTRODUCTION

The role of epigenetics has been expanded to the field of environmental toxicology to include exposure to chemical agents and pathogenesis of diseases (*Watson & Goodman, 2002*; *Szyf, 2011*). It is defined as environmental epigenetics (*Ho et al., 2012*) and provides important insights into the linkage between environmental exposure and human health based on toxicogenomic concepts (*Burris & Baccarelli, 2014*; *Reamon-Buettner, Mutschler & Borlak, 2008*).

The implication of environmental epigenetics in toxicogenomics has been demonstrated in numerous studies. It may provide the cellular and molecular signatures affected by

Corresponding author
Jae-Chun Ryu, ryujc@kist.re.kr

exposure to environmental factors and contribute to understanding epigenetic toxicological mechanisms (*Baccarelli & Bollati, 2009*). This approach is used for developing exposure biomarkers for detecting the response at low doses, early effects and elucidating the underlying modes of action for environmental disease (*McHale et al., 2010*). Therefore, it has been considered an effective strategy for toxicological risk assessment of environmental chemicals.

Exposure to a variety of environmental factors induces epigenetic alterations which emerge as key factors of numerous important cellular processes including regulation in gene expression. Also, aberrant epigenetic patterns are critical for the development of diseases and cancer progression (*Zoghbi & Beaudet, 2016*; *Kagohara et al., 2018*; *Koh & Hwang, 2019*). Furthermore, recent studies have highlighted the importance of epigenetic biomarkers such as miRNA and DNA methylation-based biomarkers. Epigenetic biomarkers are emerging as screening tools for exposure and risk assessments of environmental chemicals (*Ray, Yosim & Fry, 2014*). However, the use of epigenetic changes as a predictive exposure biomarker for exposure to environmental toxicants remains unclear. Here, we aimed to identify the epigenetic biomarkers of hexanal (hexaldehyde) for exposure and risk assessment based on the DNA methylome signature.

Hexanal is one among the aldehydes which are classified as microbial volatile organic compounds (mVOCs). mVOCs are emitted during metabolism in micro-organisms, including fungi and bacteria. It is known that mVOCs are highly abundant in the indoor environment (*Korpi, Järnberg & Pasanen, 2009*). Previous studies demonstrated that exposure to mVOCs may induce diverse adverse health effects such as irritation of the respiratory tract and eyes and inflammatory responses (*Korpi, Järnberg & Pasanen, 2009*; *Thorn & Greenman, 2012*). Of the more than 1,000 compounds of mVOCs, aldehydes are a predominant group (*Garcia-Alcega et al., 2018*). However, the toxicological data of mVOCs using omics technologies is still not well understood. We previously investigated the toxicogenomic response of hexanal, an important indoor air pollutant, using an in vitro system (*Cho et al., 2014*; *Cho et al., 2015*). In this study, we aimed to investigate the epigenetic response based on DNA methylation of hexanal exposure using the in vivo model system.

To clarify the DNA methylation networks by exposure to hexanal associated with lung toxicity, we analyzed the DNA methylation profiling of lung tissues of F-344 rats following inhalation exposure to hexanal. In the three hexanal inhalation exposure groups (600, 1,000, and 1,500 ppm), the expression of 73 methylated genes was altered and 36 dose-dependent methylated genes were also identified using a 3.0-fold change cut-off and $p$-value $< 0.05$.

To further investigate the effect of hexanal exposure on DNA methylation and gene expression profiles, we conducted an integrated analysis of the DNA methylation and mRNA expression profiles. Core anti-correlated genes which are involved in key biological processes associated with pulmonary toxicity were identified. These results provide that a novel epigenetic biomarker of exposure to hexanal and potential important quantitative biomarkers for risk assessments. This approach of DNA methylation-environmental factors may also reveals new mechanistic insights on the epigenetic actions of pulmonary toxicity.

## MATERIALS & METHODS

### Vertebrate animal study

#### Test animal

Forty male and female Fischer 344 rats of both sexes (10 rats/group), 7 weeks of age, were purchased from ORIENT BIO INC. (Seongnam, Korea). Prior to the experiment, animals were housed in stainless-steel cages (255 W × 3465 L × 3200 H mm) and acclimated for 5 days. Purification and quarantine periods were 3 or less, and during pretest and exposure periods, 2 or less were kept in stainless-steel cages. During the acclimation period, all animals are observed once a day to see clinical symptoms caused by the disease. Animals with diseases or abnormalities observed on physical examination are euthanized through $CO_2$ inhalation. Animal rooms had a 12-h light/dark cycle and controlled temperature (22 ± 3 °C) and humidity (30–70%). All animals were given a sterilized commercial pellet diet (PMI Nutrition International, USA) and sterilized water. All experimental procedures were approved by the Institutional Animal Care and Use Committee (IACUC) of Korea Institute of Toxicology (IACUC No. 1311-0301).

### Clinical, biochemical and histopathological examinations

Test animals were subjected to examine every day for any clinical, blood biochemical and histopathological symptoms and mortality. Total body weight was measured twice a week during the 4 weeks exposure period. Test animals surviving to the end of the exposure period received completed necropsy. Test animals were euthanized using isoflurane anesthesia. For autopsy animals, gross autopsy findings were observed before organ weight measurement. Whole blood (WB) was rapidly collected for blood biochemical analysis from the abdominal aorta under isoflurane. Serum was obtained from WB by centrifuging at 3,000× rpm for 10 min at room temperature and analyzed for AST (Aspartate aminotransferase), ALT (Alanine aminotransferase), ALP (Alkaline phosphatase), CK (Creatine phosphokinase), GLU (Glucose), TP (Total protein), ALB (Albumin), GLO (Globulin), A/G (Albumin/globulin ratio), BUN (Blood urea nitrogen), CREA (Creatinine), TG (Triglyceride), PL (Phospholipid), TCHO (Total cholesterol), TBIL (Total bilirubin), GGT (Gamma glutamyl transferase), Ca(Calcium), IP (Inorganic phosphorus), Cl (Chloride), Na (Sodium) and K (Potassium) using an autochemical analyzer, Toshiba 120FR NEO (Toshiba Co., Japan). The lung tissues were collected from all animals and preserved in 10% neutral buffered formalin and embedded with paraffin wax. Tissues were stained with hematoxylin and eosin (H&E) (*Cho et al., 2016*; *Cho et al., 2017*).

### Exposure design

All animal experiments were carried in accordance with relevant guidelines and regulations. Exposure experiments were designed following the OECD guideline for the testing of chemicals No. 412 ''Subacute Inhalation Toxicity'' (OECD, 2009), considering animal welfare. Hexanal vapor was generated with a bubbling generator and animals were exposed to it inside a flow-past nose-only inhalation chamber. Hexanal exposure concentrations were at target levels of 600, 1,000, and 1,500 ppm, and the control group was exposed

to filtered clean air. Grouped animals had a pre-exposure period of about 2 days before exposure began. During the pre-exposure period, holder adaptation training was performed in accordance with the standard operation procedure to reduce stress caused by non-inhalational exposure. Residual animals excluded from the test were euthanized with $CO_2$. The animals (10 rats per group) were exposed to hexanal for 4 weeks (4 h/day, 5 days/week) in the nose-only inhalation chamber. Using a GC-FID (SHIMADZU, Japan), exposure concentration of hexanal vapor was measured thrice daily. We also monitored the environment in the inhalation chamber such as chamber flow rate, temperature (°C), relative humidity (%), chamber pressure (-Pa) and oxygen concentration (%) more than 4 times during the exposure period (*Cho et al., 2016*; *Cho et al., 2017*).

## DNA preparation

Genomic DNA was isolated from the homogenized lung tissue of rats, and only the supernatant was used for extraction. DNA samples of 6 rats from each group (control, low-dose, middle-dose, and high-dose group; a total of 24 DNA samples) were used for the microarray analysis for all 40 rats used in the study. Using Qiagen's QIAamp DNA Mini kit (QIAGEN, Hilden, Germany), genomic DNA was extracted as described in our previous study (*Cho et al., 2018*). The genomic DNA purity and concentration were measured using ND-1000 spectrophotometer (NanoDrop Technologies, Wilmington, DE) and electrophoresis conducted in a 1.5% agarose gel in $1 \times$ TAE buffer (4.8 g of Tris, 1.14 mL of acetic acid, 2 mL of 0.5 M EDTA at pH 8.0, and ethidium bromide) at a constant 100 V for 15 min.

## Fragmentation of DNA

To extract only methylated DNA, the genomic DNA size should be about 200 bp to 1,000 bp. Therefore, genomic DNA was fragmented into 200 bp to 1,000 bp sections using a Sonic Dismembrator 550 (Fisher Scientific, Waltham, MA, USA) with 3 cycles comprising 4 cycles of 20 s 'ON' and 1 cycle of 20 s 'OFF'. To determine the size of the fragmented DNA, agarose gel electrophoresis and ethidium bromide staining were performed using DNA size markers of 500–10,000 base pairs in size.

## Methylated DNA immunoprecipitation (MeDIP)

As described in our previous study (*Cho et al., 2018*), MeDIP was performed with MethylMiner Methylated DNA Enrichment Kit (Invitrogen, Carlsbad, CA, USA) according to the manufacturer's instructions. Fragmented DNA 1 μg and untreated control DNA (Input) 3 μg were used for quality and labelling procedures. Briefly, Dynabeads M-280 Streptavidin 10 μl was combined with 7 μl of MBD (methyl-CpG binding domain)-Biotin Protein. The MBD-magnetic bead conjugates were washed thrice and resuspended in 1 volume of 1X bind/wash buffer. The capture reaction was conducted by adding of 1 μg sonicated DNA to the MBD magnetic beads on a rotating mixer for 1 h at room temperature. Next, the beads were washed three times with $1 \times$ bind/wash buffer. The methylated DNA was eluted as a single fraction with a high-salt elution buffer (2,000 mM NaCl). Consequently, each fraction was concentrated by ethanol precipitation using 1 μL glycogen (20 μg/μL), 1/10th volume of 3 M sodium acetate (pH 5.2), and two

volumes of 100% ethanol, and then resuspended in 60 µL of DNase-free water. The eluted methylated DNA immunoprecipitation samples were stored at −20 °C until further use. This experiment protocol was referred from out previous research in *Cho et al. (2018)*.

## Epigenome-wide DNA methylation

First, whole genome amplification kit (GenomePlex Complete Whole Genome Amplification Kit, SIGMA-ALDRICH, USA) was used to amplify DNA and methylated immunoprecipitation (IP) samples according to the manufacturer's instructions. The amplified samples were purified using the QIAQuick PCR clean-up kit (QIAGEN, Hilden, Germany). The amplified DNA and 4 µg of the methylated IP sample were labeled using the Bioprime labeling kit from Invitrogen according to the manufacturer's instructions. The IP sample was labeled with Cy5-dUTP and the input DNA sample was labeled with Cy3-dUTP and 50 µl of master mix(dNTPs-dATP, dGTP, dCTP; 120 µM, dTTP; 60 µM, Cy5-dUTP or Cy3-dUTP; 60 µM). After labeling the sample, the concentration was measured using an ND-1000 spectrometer (NanoDrop Technologies, Inc., Wilmington, DE).

Second, After checking labeling efficiency, each 2.5ug to 5ug of cyanine 3-labeled and cyanine 5-labeled DNA target were mixed and then resuspended with 2X hybridization buffer, Cot-1 DNA, and Agilent 10X blocking agent, and de-ionized formamide. Before hybridization to the array, the 260ul hybridization mixtures were denatured at 95 °C for 3min and incubated at 37 °C for 30 min. The hybridization mixtures were was centrifuged at 17,900× g for 1min and directly pipetted onto the Customized Rat Methylation Microarray (400 K). The arrays hybridized at 65oC for 40 h using Agilent Hybridization oven (Agilent Technology, USA). The hybridized microarrays were washed as the manufacturer's washing protocol (Agilent Technology, Santa Clara, CA, USA).

Third, after washing, hybridization images on the slides were scanned using the Agilent DNA microarray scanner (Agilent Technologies) and signals were extracted from each probe using Agilent Feature Extraction software (v10.7.3.1). All data were normalized using Agilent's Workbench software v7.0 according to the manufacturer's instructions (Agilent Technologies). The background-corrected intensity data were normalized with blank subtraction followed by intra-array LOWESS normalization. The peak detection was performed with Pre-defined Peak Shape detection v2.0 with a *p*-value <0.01 for non-parametric test and a peak-score >5 for EVD-based score. The data were normalized by dividing the average of the signal intensity of the exposed group by the normalized average of the control group. The differentially methylated probes were selected using the 3.0-fold change cutoff and *p*-value <0.05. For reference, the intensity dependent normalization is a technique that is used to eliminate dye-related artifacts in two-color experiments that cause the cy5/cy3 ratio to be affected by the total intensity of the spot. This normalization process attempts to correct for artifacts caused by non-linear rates of dye incorporation as well as inconsistencies in the relative fluorescence intensity between some red and green dyes.

## Integrating DNA methylation and gene expression

To identify the anti-correlated methylated genes, we conducted a comparative analysis of DNA methylation and mRNA expression patterns using GeneSpring GX. mRNA profiles

from the hexanal-exposed rats were obtained from our previous study (*Cho et al., 2017*). We used Pearson's correlation analysis, the most appropriate statistical coefficient for a small number of measures, to estimate the degree of anti-correlation (e.g., hyper methylation vs. down-regulated c mRNA expression or vice versa) between any putative pairs of DNA methylation and mRNA. The raw data are available from the NCBI GEO under accession number GSE60118. We considered the methylated genes with methylation differences of at least 3.0-fold and mRNA expression differences of at least 1.5-fold on $p$-value <0.05.

## DAVID functional enrichment analysis

Using the DAVID functional annotation bioinformatics tool, we performed GO enrichment analyses to understand biological functions associated with hexanal exposure. It was used to determine significant biological pathways for anti-correlated target genes between DNA methylation and mRNA expression associated with hexanal exposure. Fisher's exact test was used to detect significant enrichment of pathways, and the resulting $p$-value were adjusted using the Benjamini–Hochberg algorithm.

## Statistical analysis

In all cases, the differences between the control and exposure group were evaluated using the unpaired $t$-test. The $p$-value criterion was set at $p$-value <0.05 as the level of statistical significance.

## Animal ethics

The experiment protocol was authorized by the Institutional Animal Care and Use Committee of Korea Institute of Toxicology (IACUC No. 1311-0301).

# RESULTS

## Monitoring of inhalation exposure concentration, environmental conditions, and histopathologic alterations

As mentioned in our previous studies (*Cho et al., 2016*; *Cho et al., 2017*), inhalation hexanal exposure concentrations were monitored in rats using online gas chromatography (GC) every 10 min during the exposure period. SPF (Specific-pathogen-free) Fischer-344 derived (CRL:CD) rats of both sexes were used at the age of 7 weeks (n=10/group). The average exposure concentrations were 646.03 ($\pm$ 80.06; low-dose), 999.06 ($\pm$ 162.08; middle-dose), and 1,525.31 ($\pm$ 199.02; high- dose) ppm. The conditions of the inhalation chamber such as temperature, relative humidity, chamber pressure, and oxygen concentration were also measured (*Cho et al., 2016*).

Compared with the control group, no significant body weight, organ weight and histopathologic alterations were observed after 4 weeks of hexanal exposure (*Cho et al., 2017*). In middle-dose group, increased total bilirubin compared to control group in the male rats and decreased total protein, albumin and triglyceride in the female rats were identified. These results showed no significant dose-dependent changes related to hexanal exposure (*Cho et al., 2017*). Therefore, to predict the potential adverse health effects of hexanal exposure we aimed to identify the hexnal-associated genetic and epigenetic
alterations using microarray-based mRNA and DNA methylation to address the molecular basis of hexanal exposure relevant to respiratory system.

## DNA methylation pattern after hexanal exposure

Aberrant DNA methylation has been linked to the abnormalities or disorders that induced by environmental stressors including environmental chemicals (*Kubota, 2016*). Therefore, the framework of epigenome for environmental risk assessment has been rapidly developed. First, we extracted from rats exposed to hexanal of three concentrations (Low dose, 600 ppm; Middle dose, 1,000 ppm; High dose, 1,500 ppm), and then genomic DNA using sonication to extract only methylated DNA using immunoprecipitation. The cleaved methylated DNA was confirmed using gel electrophoresis, and as a result, it was confirmed that the DNA of all groups was sheared to about 150 bp to 500 bp, so that the optimal DNA for immunoprecipitation was secured (Fig. S1). After methylated DNA was extracted from aldehyde-exposed rat lung tissues through methylated DNA immunoprecipitation, the concentration was measured and the quantitative analysis of methylated DNA was performed through gel electrophoresis. As a result, it was confirmed that the concentration and state of methylated DNA are suitable for DNA methylation microarray (Fig. S2). In the current study, using a custom-designed Agilent 400K CpG methylation microarray, we investigated DNA methylation profiles in CpG islands gene promoter sequences of hexanal-exposed lung tissues of F344 rats and compared with those from rats exposed to clean filtered air (control group) (n=6/group). For reference, the DNA Methylation Microarray are designed to interrogate known CpG islands and related sites. It is designed for analysis of methylated DNA derived from affinity-based isolation methods including methylated DNA immunoprecipitation (MeDIP). We analyzed methylation patterns for approximately 389,347 probes on the arrays. Compared with the control group, all three hexanal-exposed groups showed distinctly different methylation patterns (Fig. 1). The data is the averaged signal that is acquired from normalizing the signal intensity by dividing the average of the signal intensity of the control group. In the low dose exposure group, 661 methylated sites and 571 differentially methylated genes (hyper-methylated: 464, hypo-methylated: 107) were identified. In the middle dose exposure group, 4,181 methylated sites and 3,268 differentially methylated genes (hyper-methylated: 2,513, hypo-methylated: 755) were identified, and 11,744 methylated sites and 7,477 differentially methylated genes (hyper-methylated: 4,851, hypo-methylated: 2,662) were identified in the high dose exposure group. In all groups change was noted at $\geq$ 3.0-fold change, *p*-value <0.05. Overall, the methylation sites increased as the exposure concentration increased. (Table 1).

Among these differentially methylated sites and genes, 79 sites and 73 genes (hyper-methylated: 69, hypo-methylated: 4) showed commonly methylated expression patterns in the three hexanal exposure groups (Fig. 2, Table 2). Furthermore, we identified 36 dose-dependent methylated genes (34 hyper-methylated and 2 hypo-methylated) in the common methylated genes of three hexanal exposure group using line-plot analysis (Fig. 3A, Table 3). The dose-dependent genes are illustrated as a heatmap (Fig. 3B). These dose–response relationships have the potential to serve as quantitative epigenetic

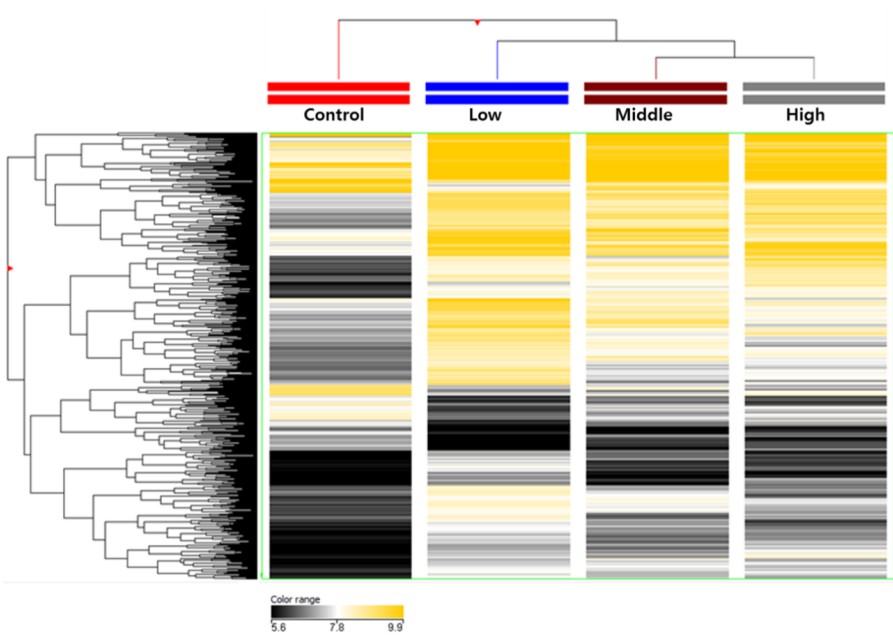

**Figure 1** **Total DNA methylation expression profiles of hexanal exposed in F-344 rats (n = 6/group).** Two-dimensional diagram of the characteristic expression profiles of 389,247 classifier methylation probes. Rows (*y*-axis) represent the intensity of the DNA methylation probes and columns (*x*-axis) represent the different experimental conditions. Color intensity reflects differences in expression between sample DNA and reference DNA.

**Table 1** **The DNA methylated sites and regulated target genes in three hexanal exposure group.**

| Exposure dose | Methylated sites | Regulated target genes |
|---|---|---|
| Low dose (600 ppm) | 661 | 571 |
| Middle dose (1,000 ppm) | 4,181 | 3,268 |
| High dose (1,500 ppm) | 11,744 | 7,477 |

biomarkers of hexanal exposure. Raw data are available online at Gene Expression Omnibus (GEO accession number GSE129313).

## Gene expression profiles induced by hexanal exposure

To investigate the gene expression signatures response to hexanal inhalation exposure, we previously investigated the gene expression profiling of lung tissues of hexanal-exposed F344 rats using the Rat Oligo Microarray (44 K). The raw data are available at GEO/NCBI GSE 60118. The gene expression profiles were analyzed by comparing them to the control group using 1.5-fold change and unpaired *t*-test *p* value <0.05 as statistical significance (Table 4). In the previous study, we identified hexanal specific genes that were involved in diverse biological processes including apoptosis, cell proliferation, and mitogen-activated protein kinase (MAPK) cascade. These genes were also associated with disease such as respiratory and nervous system diseases (*Cho et al., 2017*). It suggests that hexanal exposure

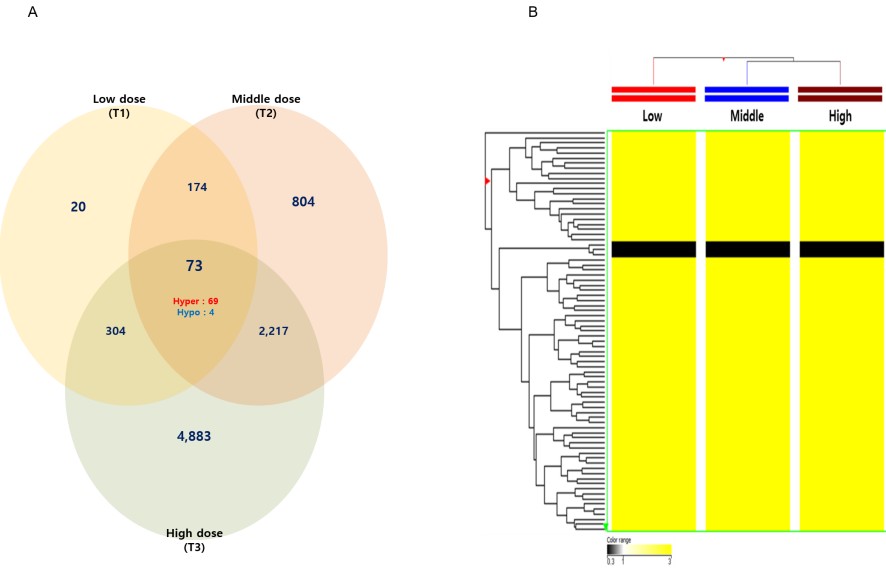

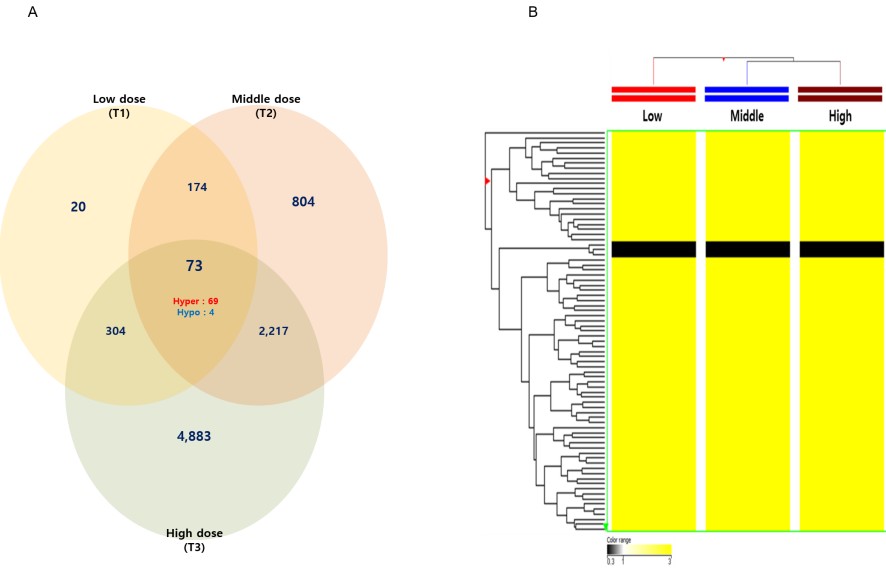

**Figure 2** **The Venn diagram and hierarchical clustering image of hexanal specific methylated DNA.**
(A) The Venn diagram and (B) hierarchical clustering image shows that 73 methylated DNA that commonly altered their expression are identified in three dose—T1(600 ppm), T2(1,000 ppm) and T3(1,500 ppm) with a fold-change ≥3.0-fold and $p$-value < 0.05 compared to the control group (Filtered air) (Yellow: hypermethylation; Black: hypomethylation).

may have potential adverse health effects on humans. Therefore, we aimed to analyze DNA methylation signatures in hexanal- exposed F344 rats to understand the epigenetic effects of hexanal exposure.

## Comparative analysis of DNA methylation and mRNA expression profiles

DNA methylation was involved in transcriptional regulation and gene activity. Promoter hyper-methylation can leads to silencing of gene expression, whereas hypo-methylation can leads to gene activation. The investigation of the implication of DNA methylation in the regulation of gene expression and identification of key genes that regulated by both DNA methylation and gene expression using integrative analysis is important. Therefore, we conducted an integrated analysis of DNA methylation (Table 1) and mRNA expression data (Table 4). As shown in Table 5, we identified the hyper-methylated vs. down-regulated genes and hypo-methylated vs. up-regulated genes in the hexanal exposure groups. These results suggest potential core DNA methylation-based epigenetic biomarkers for exposure/risk assessment of hexanal.

## Gene Ontology (GO) analysis of putative DNA methylation biomarkers of hexanal

We next investigated the relevant molecular and cellular processes controlled by hexanal exposure-specific inversely correlated target genes based on GO biological processes terms using the DAVID bioinformatics tool (Table 6). The key GO terms were related to the lactation (GO:0007595), skeletal muscle cell differentiation (GO:0035914), Positive

**Table 2** The list of 73 methylated target genes that commonly altered their expression in three hexanal exposure group.

| Probe ID | Annotation | Fold change | | | Regulation |
|---|---|---|---|---|---|
| | | Low dose (T1) | Middle dose (T2) | High dose (T3) | |
| RP14104253 | Kcng1 | 0.31 | 0.27 | 0.33 | Down |
| RP14316148 | Pcca | 0.28 | 0.28 | 0.23 | Down |
| RP14196310 | Prkcsh | 0.30 | 0.29 | 0.24 | Down |
| RP14232470 | Sstr5 | 0.27 | 0.29 | 0.18 | Down |
| RP14072666 | Adh4 | 3.69 | 5.67 | 6.99 | Up |
| RP14052111 | Ankrd34b | 3.34 | 5.67 | 3.86 | Up |
| RP14347714 | Atp9b | 3.82 | 5.05 | 3.70 | Up |
| RP14104488 | Bcas1 | 4.10 | 3.07 | 6.10 | Up |
| RP14068918 | Capza1 | 4.21 | 3.73 | 5.01 | Up |
| RP14275323 | Ccl24 | 3.38 | 3.97 | 3.64 | Up |
| RP14132603 | Ccnc | 3.17 | 4.21 | 9.26 | Up |
| RP14271990 | Ccz1 | 3.69 | 3.98 | 5.10 | Up |
| RP14133935 | Chmp5 | 3.55 | 3.30 | 3.65 | Up |
| RP14222274 | Crygb | 3.80 | 5.65 | 6.50 | Up |
| RP14271525 | Cyp3a23/3a1 | 3.04 | 4.79 | 4.29 | Up |
| RP14329375 | Ddx41 | 3.38 | 6.09 | 6.58 | Up |
| RP14200524 | Dpagt1 | 3.15 | 3.10 | 4.53 | Up |
| RP14344056 | Dtwd2 | 3.03 | 6.60 | 10.47 | Up |
| RP14214176 | Eif1b | 4.11 | 4.07 | 12.6.37 | Up |
| RP14372792 | Fam228a | 3.13 | 4.24 | 3.59 | Up |
| RP14266770 | Fgf12 | 4.25 | 3.27 | 3.56 | Up |
| RP14338669 | Fundc2 | 4.84 | 8.01 | 10.45 | Up |
| RP14253846 | G6pc3 | 4.79 | 3.86 | 3.74 | Up |
| RP14008399 | Gltscr2 | 3.38 | 4.32 | 4.30 | Up |
| RP14108468 | Hgf | 3.15 | 3.74 | 6.07 | Up |
| RP14139108 | Hook1 | 4.25 | 3.50 | 5.47 | Up |
| RP14301839 | Inpp5j | 3.49 | 7.37 | 8.39 | Up |
| RP14183524 | Jrk | 3.15 | 4.97 | 7.04 | Up |
| RP14214724 | Kif15 | 3.23 | 4.71 | 4.62 | Up |
| RP14016091 | Klk1c9 | 3.21 | 3.21 | 4.89 | Up |
| RP14248629 | LOC303448 | 3.74 | 5.28 | 4.63 | Up |
| RP14343955 | LOC317165 | 3.44 | 4.13 | 3.41 | Up |
| RP14299892 | Lyar | 3.51 | 4.27 | 3.36 | Up |
| RP14208423 | Mrap2 | 3.18 | 5.16 | 4.73 | Up |
| RP14207074 | Myo5a | 3.37 | 4.16 | 4.77 | Up |
| RP14169885 | Naca | 4.46 | 5.24 | 6.78 | Up |
| RP14301172 | Nelfa | 4.68 | 3.28 | 4.22 | Up |
| RP14171345 | Olr1049 | 3.31 | 3.21 | 4.30 | Up |
| RP14175180 | Olr1084 | 3.14 | 3.22 | 3.42 | Up |
**Table 2** (*continued*)

| Probe ID | Annotation | Fold change | | | Regulation |
|---|---|---|---|---|---|
| | | Low dose (T1) | Middle dose (T2) | High dose (T3) | |
| RP14175183 | Olr1085 | 3.97 | 4.65 | 4.98 | Up |
| RP14199556 | Olr1328 | 3.99 | 3.65 | 5.69 | Up |
| RP14235002 | Olr1389 | 3.38 | 3.57 | 5.12 | Up |
| RP14307926 | Olr1624 | 9.62 | 24.83 | 12.97 | Up |
| RP14359583 | Olr1696 | 3.46 | 5.37 | 6.49 | Up |
| RP14359635 | Olr1701 | 4.19 | 3.05 | 5.11 | Up |
| RP14080399 | Olr407 | 3.43 | 3.79 | 4.18 | Up |
| RP14085578 | Olr448 | 3.37 | 4.72 | 4.11 | Up |
| RP14006401 | Olr5 | 3.17 | 3.51 | 6.25 | Up |
| RP14086255 | Olr500 | 3.90 | 8.02 | 9.10 | Up |
| RP14342389 | Pcdhb12 | 3.26 | 3.18 | 3.44 | Up |
| RP14280875 | Pitpnb | 3.81 | 5.59 | 4.57 | Up |
| RP14279738 | Pla2g1b | 3.73 | 4.07 | 4.56 | Up |
| RP14131422 | Plag1 | 3.24 | 3.72 | 4.16 | Up |
| RP14086989 | Pramel6 | 3.75 | 4.52 | 5.44 | Up |
| RP14066541 | Prune | 3.03 | 4.54 | 7.46 | Up |
| RP14335189 | Psma2 | 3.35 | 3.78 | 3.78 | Up |
| RP14055154 | RGD1306227 | 3.18 | 4.06 | 4.69 | Up |
| RP14108982 | RGD1564345 | 3.18 | 6.32 | 8.11 | Up |
| RP14374247 | Rhox3 | 3.24 | 4.87 | 3.71 | Up |
| RP14029004 | rnf141 | 3.05 | 8.04 | 5.87 | Up |
| RP14212856 | Rtp3 | 3.35 | 6.35 | 5.19 | Up |
| RP14076254 | Sdccag3 | 3.69 | 4.58 | 5.01 | Up |
| RP14344660 | Slc12a2 | 3.32 | 4.88 | 6.02 | Up |
| RP14170272 | Slc39a5 | 5.49 | 6.28 | 7.65 | Up |
| RP14130014 | Slco1a2 | 3.20 | 3.40 | 3.96 | Up |
| RP14165691 | Slirp | 3.82 | 4.60 | 4.47 | Up |
| RP14049010 | Taf5 | 3.19 | 4.32 | 3.27 | Up |
| RP14053110 | Tmem174 | 5.15 | 4.73 | 4.69 | Up |
| RP14211530 | Traip | 3.45 | 5.65 | 6.02 | Up |
| RP14096780 | Trmt6 | 4.06 | 3.93 | 5.93 | Up |
| RP14013877 | Tyrobp | 3.60 | 4.79 | 3.66 | Up |
| RP14288987 | Usf1 | 3.09 | 3.16 | 3.85 | Up |
| RP14237317 | Zfp672 | 3.69 | 5.22 | 4.65 | Up |

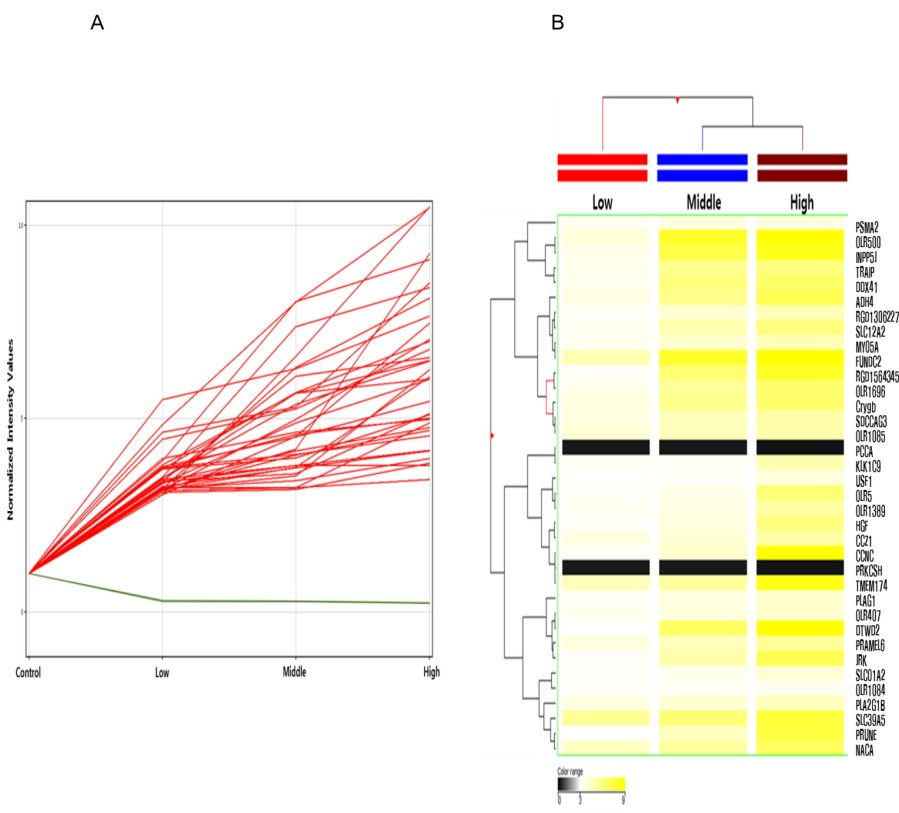

**Figure 3  Line plot and heatmap of dose-dependent response of methylated genes by hexanal exposure.**
(A) The line plot shows dose-dependent response of methylated genes by hexanal exposure. Each line of the plot represents the normalized intensity values by the control group shown on the *x*-axis. The y-axis has a log2 scale. (B) The heatmap of 36 dose-dependent methylated genes by hexanal exposure.

regulation of synapse assembly (GO:0051965), sodium ion transport (GO:0006814), and regulation of tumor necrosis factor production (GO:0032680). These results indicated that putative epigenetic biomarkers of hexanal are involved in skeletal muscle cell differentiation, synapse assembly, and TNF production. Further studies are necessary to determine hexanal-induced toxicological mechanisms based on functional enrichment analysis.

## DISCUSSION

Traditional toxicity testing depends on animal testing to investigate the risk of chemicals to human health. It requires several animals, high investment, and a significant amount of time. Additionally, it should consider ethical treatment of animals and their welfare. Therefore, this approach is insufficient to handle risk assessments of the large number of chemicals in the environment (*Chen et al., 2012*; *North & Vulpe, 2010*), and novel strategies for toxicological risk assessment of environmental chemicals are necessary.

In response to these challenges, the field of toxicogenomics has been established and developed rapidly for risk assessments. Toxicogenomics includes high-throughput technologies such as transcriptomics, proteomics, and metabolomics for predictive

**Table 3  The dose-dependent methylated target genes in three hexanal exposure group.**

| Probe ID | Annotation | Fold change | | | Regulation |
|---|---|---|---|---|---|
| | | Low dose (600 ppm) | Middle dose (1,000 ppm) | High dose (1,500 ppm) | |
| RP14196310 | Prkcsh | 0.30 | 0.29 | 0.24 | Down |
| RP14316148 | Pcca | 0.28 | 0.28 | 0.23 | Down |
| RP14006401 | Olr5 | 3.17 | 3.51 | 6.25 | Up |
| RP14016091 | Klk1c9 | 3.21 | 3.21 | 4.89 | Up |
| RP14053108 | Tmem174 | 4.64 | 5.29 | 8.50 | Up |
| RP14055154 | RGD1306227 | 3.18 | 4.06 | 4.69 | Up |
| RP14066541 | Prune | 3.03 | 4.54 | 7.46 | Up |
| RP14072666 | Adh4 | 3.69 | 5.67 | 6.99 | Up |
| RP14076254 | Sdccag3 | 3.69 | 4.58 | 5.01 | Up |
| RP14080399 | Olr407 | 3.43 | 3.79 | 4.18 | Up |
| RP14086255 | Olr500 | 3.90 | 8.02 | 9.10 | Up |
| RP14086989 | Pramel6 | 3.75 | 4.52 | 5.44 | Up |
| RP14108468 | Hgf | 3.15 | 3.74 | 6.07 | Up |
| RP14108982 | RGD1564345 | 3.18 | 6.32 | 8.11 | Up |
| RP14130014 | Slco1a2 | 3.20 | 3.40 | 3.96 | Up |
| RP14131422 | Plag1 | 3.24 | 3.72 | 4.16 | Up |
| RP14132603 | Ccnc | 3.17 | 4.21 | 9.26 | Up |
| RP14169885 | Naca | 4.46 | 5.24 | 6.78 | Up |
| RP14170272 | Slc39a5 | 5.49 | 6.28 | 7.65 | Up |
| RP14175180 | Olr1084 | 3.14 | 3.22 | 3.42 | Up |
| RP14175183 | Olr1085 | 3.97 | 4.65 | 4.98 | Up |
| RP14183524 | Jrk | 3.15 | 4.97 | 7.04 | Up |
| RP14207074 | Myo5a | 3.37 | 4.16 | 4.77 | Up |
| RP14211530 | Traip | 3.45 | 5.65 | 6.02 | Up |
| RP14222274 | Crygb | 3.80 | 5.65 | 6.50 | Up |
| RP14235002 | Olr1389 | 3.38 | 3.57 | 5.12 | Up |
| RP14271990 | Ccz1 | 3.69 | 3.98 | 5.10 | Up |
| RP14279738 | Pla2g1b | 3.73 | 4.07 | 4.56 | Up |
| RP14288987 | Usf1 | 3.09 | 3.16 | 3.85 | Up |
| RP14301839 | Inpp5j | 3.49 | 7.37 | 8.39 | Up |
| RP14329375 | Ddx41 | 3.38 | 6.09 | 6.58 | Up |
| RP14335189 | Psma2 | 3.35 | 3.78 | 3.78 | Up |
| RP14338669 | Fundc2 | 4.84 | 8.01 | 10.45 | Up |
| RP14344056 | Dtwd2 | 3.03 | 6.60 | 10.47 | Up |
| RP14344660 | Slc12a2 | 3.32 | 4.88 | 6.02 | Up |
| RP14359583 | Olr1696 | 3.46 | 5.37 | 6.49 | Up |

toxicology and risk assessment (*Hamadeh et al., 2002*). Currently, an integrated framework for multi-omics has been proposed. It provides insight into the mode of action of environmental toxicants and helps in understanding the underlying mechanisms of toxicants and adverse outcome pathways (AOPs) (*Williams, Mirbahai & Chipman, 2014*).

**Table 4** The number of differentially expressed genes (DEGs) in three hexanal exposure group with 1.5-fold change cutoff and *p*-value < 0.05).

| | Up-regulated genes | Down-regulated genes | Total genes |
|---|---|---|---|
| **Low dose (600 ppm)** | 73 | 570 | 643 |
| **Middle dose (1,000 ppm)** | 600 | 211 | 811 |
| **High dose (1,500 ppm)** | 359 | 210 | 569 |

**Table 5** GO (Gene Ontology) analysis of target genes show that the key biological processes under hexanal inhalation exposure (*p*-value < 0.05).

| GO Accession No. | GO Term | Count | *p*-value | Genes |
|---|---|---|---|---|
| GO:0007595 | Lactation | 4 | 0.001 | NM_013197 (ALAS2), NM_012630 (PRLR), NM_001012027 (SERPINC1), NM_001013248 (FOXB1) |
| GO:0035914 | Skeletal muscle cell differentiation | 3 | 0.015 | NM_017259 (BTG2), NM_024388 (NR4A1), NM_013220 (ANKRD1) |
| GO:0051965 | Positive regulation of synapse assembly | 3 | 0.023 | NM_134376 (CLSTN3), NM_001109430 (LRTM2), NM_012892 (ASIC2) |
| GO:0006814 | Sodium ion transport | 3 | 0.037 | NM_001113335 (SLC9A2), NM_012892 (ASIC2), NM_001109385 (SLC9B2) |
| GO:0009612 | Response to mechanical stimulus | 3 | 0.037 | NM_017259 (BTG2), NM_012892 (ASIC2), NM_021836 (JUNB) |
| GO:0032680 | Regulation of tumor necrosis factor production | 2 | 0.025 | NM_133290 (ZFP36), NM_001106864 (LTF) |
| GO:0060213 | Positive regulation of nuclear-transcribed mRNA poly(A) tail shortening | 2 | 0.025 | NM_133290 (ZFP36), NM_017259 (BTG2) |

**Table 6** The number of correlated target genes between DNA methylation and mRNA expression by hexanal exposure (*p*-value < 0.05).

| | Hyper-methylated vs. Down-regulated | Hypo-methylated vs. Up-regulated |
|---|---|---|
| Low dose (600 ppm) | 7 | 0 |
| Middle dose (1,000 ppm) | 24 | 25 |
| High dose (1,500 ppm) | 44 | 28 |

In contrast to traditional toxicity methods, it is possible to also identify multiple-response and endpoints using toxicogenomics.

Toxicogenomics study has developed rapidly with microarray and next generation sequencing technologies. The microarray technology was proposed in the 1990s (*Chen et al., 2012*). It is a powerful tool for evaluating the effect of environmental chemicals on human health, providing valuable genomic information for identifying biomarkers related to occupational exposure and disease prognosis (*Jung et al., 2017*; *Gwinn & Weston, 2008*; *Kim et al., 2016*). It allows simultaneous screening of the expression levels of thousands of genes exposed to environmental toxicants based on omics tools. Therefore, toxicogenomics

has been considered as a new toxicology paradigm for risk assessment and prediction of exposure and risk of environmental chemicals.

One of the epigenome studies demonstrated that DNA methylation has an important role in the regulation of gene expression and epigenetic phenotype variation (*Hong et al., 2018*) leading to insights into the development of diseases associated with environmental risk assessment (*Ray, Yosim & Fry, 2014*; *Conerly & Grady, 2010*). Generally, the expression patterns of DNA methylation are altered by environmental factors, including environmental chemicals, air pollution, and nonchemical stressors. Moreover, it has been linked to levels of health, disease susceptibility, and disease development (*Martin & Fry, 2018*). Therefore, epigenetic modifications can be novel exposure biomarkers of the diseases related to environmental factors.

To investigate the epigenetics actions of hexanal associated with lung toxicity, we aimed to identify epigenetic biomarkers based on DNA methylation. As major component of indoor air pollutants, we previously analyzed the transcriptome profiles of hexanal using in vitro and in vivo models (*Cho et al., 2014*; *Cho et al., 2017*). And we also analyzed the methylation profiles of seven aldehydes (propanal, butanal, pentanal, hexanal, heptanal, octanal, and nonanal) exposed human lung epithelial cell, A549, to investigate the aldehydes exposure and epigenetic alterations based on DNA methylation (*Cho et al., 2018*). Here, we proposed three steps of DNA methylome analysis of hexanal exposure using the in vivo model. First, we identified the differentially methylated genes of hexanal exposure showing a 3.0-fold-change ($p < 0.05$). Of the 389,347 probes on the customized rat 400K CpG methylation microarray, the methylated genes identified showed significant expression changes in the three hexanal exposure groups (low dose, middle dose, and high dose) compared to the control group. Among the differentially methylated genes, we identified commonly methylated genes and dose-dependent methylated patterns, which provided significant novel epigenetic biomarkers of hexanal exposure. These methylated genes were involved in chemical stimulus associated with olfactory receptor activity (OLR1696, OLR500, OLR5, OLR407, OLR1085, OLR1084, OLR1389), insulin stimulus (PLA2G1B, MYO5A, USF1) and negative regulation of peptidyl-serine phosphorylation (HGF, INPP5J). The follow-up studies will be necessary to address a pulmonary toxicological mechanisms associated with hexanal exposure. Also, the dose–response relationship plays essential role in the field of toxicology, it provides the determination of threshold for toxic effect and better understanding of network for exposure-human health (*Tsatsakis et al., 2018*).

Second, we analyzed the transcriptome profiles of hexanal exposure in F344 rats to investigate the hexanal-characterized genes and environmental chemical-gene interactions based on toxicogenomics (*Cho et al., 2017*). Third, we conducted the comparative analysis of genome-wide DNA methylome and transcriptome in the hexanal- exposed F344 rats. It is well known that the DNA methylation is associated with gene expression. DNA hypermethylation results in gene silencing and hypomethylation leads to elevated transcription (*Li et al., 2017*). The identification of key genes that regulated by both DNA methylation and mRNA expression system via integrative analysis is necessary. Therefore, we aimed to identify the novel biomarkers that anti-correlated between DNA methylation and gene expression. Together, these processes can serve to determine the important

framework for environmental epigenetics in exposure/risk assessment and it allows the identification of the critical bridging epigenetic biomarkers of hexanal. Further biomarker validation and developing studies are necessary to explore the specificity, sensitivity and implications of these biomarkers. And then, it is predicted that this epigenetic biomarkers can be used to determine whether exposure to hexanal and to determine the cause of environmental diseases.

In vivo models including rat model are essential for evaluating the toxicity of inhaled factors for the risk assessment on human health. It is fundamental for understanding the mammalian system including human biology at molecular level. Therefore, we used the F344 rat models to evaluate the pulmonary toxicity of hexanal associated with human adverse health effects. In this study, the analyzed DNA methylated genes at CpG islands were conserved in human. It has orthologs between rat and human.

Most of aldehydes inhalation toxicity research has progressed extensively on formaldehyde and acetaldehyde, which are classified as Group 1 carcinogens by IARC (International Agency for Research on Cancer). However, other aldehydes such as hexanal toxicological data are relatively insufficient for risk assessment. Therefore, we aimed to investigate the inhalation toxicity of hexanal using F344 rats. For reference, in this study, hexanal exposure doses (low dose,600 ppm; middle dose, 1,000 ppm; and high dose, 1,500 ppm) were selected based on the $LC_{Lo}$ (Lowest Lethal Concentration; 2,000 ppm/4hr) of hexanal using nose-only inhalation chamber. These exposure dose levels that are much higher than actually exposed levels in environment. Since the VOCs are typically exposed to low levels for long-term, we determined the hexanal exposure doses higher than the actual exposure levels to investigate the clear implications for human health.

Using the DAVID functional annotation bioinformatics tool, GO analysis was also performed. GO enrichment analysis demonstrated that cell differentiation of skeletal muscle cells, regulation of synapse assembly and regulation of TNF production are involved in major biological process associated with hexanal exposure. Among them, BTG2, ZFP36 and ASIC2 were commonly involved in hexanal related biological processes such as skeletal muscle cell differentiation and regulation of nuclear-transcribed mRNA poly (A) tail shortening. BTG2 (BTG anti-proliferation factor 2) has important roles in control of cell growth, cell differentiation, apoptosis and transcriptional regulation. Moreover, it is involved in tumor progression in response to a variety of stressors, steroid hormones and growth factors (*Yuniati et al., 2019*). ZFP36 (Zinc finger protein 36 homolog; also known as Tristetraprolin) plays role in regulation of TNF-$\alpha$ (Tumor necrosis factor-alpha) expression which is a pro-inflammatory mediator (*Zhao et al., 2016*). Since we identified that the relationship between the TNF regulation and hexanal exposure using GO analysis, we considered that further research on inflammatory mechanisms via TNF associated with ZFP36 expression is required. ASIC2 (Acid sensing ion channel subunit 2) is expressed in several systems including peripheral and central nervous system as mechanoreceptor and acid receptor (*Kikuchi et al., 2008*). Recent studies demonstrated that ASIC2 may lead to increase the pulmonary vascular resistance and possibility of hypoxic pulmonary hypertension (*Detweiler et al., 2019*).

These results reflect that hexanal exposure may affect skeletal muscle and neuronal system as well as respiratory system. Further validation of key toxicological mechanisms induced by hexanal exposure such as pulmonary inflammation via TNF signaling pathway is required.

## CONCLUSIONS

Taken together, this study demonstrated the characteristic methylated profiles by hexanal inhalation exposure system using DNA methylome analysis in an in vivo model. By integrating DNA methylation and mRNA expression profiles, target genes were identified. These genes could be valuable epigenetic biomarkers to distinguish exposure to hexanal and to determine the DNA methylome responses to hexanal exposure in the environment and to predict the underlying mechanisms of hexanal exposure associated with pulmonary toxicity. Further studies on these methylated signatures are required to provide insights into the molecular toxicological mechanisms activated by hexanal exposure.

### Funding

This research was supported by the Korea Research Foundation grants from Korea Ministry of Environment as ''The Ecoinnovation Project (412-111-010)'', and the KIST Program to Jae-Chun Ryu of the Republic of Korea. The funders had no role in study design, data collection and analysis, decision to publish, or preparation of the manuscript.

### Grant Disclosures

The following grant information was disclosed by the authors:
Korea Research Foundation grants from Korea Ministry of Environment as ''The Ecoinnovation Project (412-111-010)''.
KIST Program to Jae-Chun Ryu of the Republic of Korea.

### Competing Interests

The authors declare there are no competing interests.

### Author Contributions

- Yoon Cho conceived and designed the experiments, performed the experiments, analyzed the data, prepared figures and/or tables, authored or reviewed drafts of the paper, and approved the final draft.
- Mi-Kyung Song and Jae-Chun Ryu conceived and designed the experiments, analyzed the data, authored or reviewed drafts of the paper, and approved the final draft.

### Animal Ethics

The following information was supplied relating to ethical approvals (i.e., approving body and any reference numbers):

The experiment protocol was authorized by the Institutional Animal Care and Use Committee of Korea Institute of Toxicology (IACUC No. 1311-0301). Exposure

experiments were designed following the OECD guideline for the testing of chemicals No. 412 "Subacute Inhalation Toxicity".

## Data Availability

Raw data are available online at the Gene Expression Omnibus: GSE129313.

## Supplemental Information

Supplemental information for this article can be found online at http://dx.doi.org/10.7717/peerj.10779#supplemental-information.

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
