# Peer review of "DNA methylome signatures as epigenetic biomarkers of hexanal associated with lung toxicity"

_PeerJ, doi:10.7717/peerj.10779_

## Round 0.1 · original submission · Major Revisions

The referees have carefully reviewed your manuscript. Please address them and submit your revised manuscript.

Reviewer 1 ·

Basic reporting

no comment

Experimental design

no comment

Validity of the findings

no comment

Additional comments

Exposure to hexanal is an interesting issue and the use of epigenetics as a potential marker is intriguing even if the paper is only observational and limited to an in silico analysis of pathways and targets epigenetically altered.
Some minor revisions:
The authors should better explain the reasons for the three dosages: the definition of low, medium, high or even the dosage itself is not clear if not contextualized, if not related to dosages which are representative for humans, for instance. In this light, the authors should explain how, or if, their model fits into potential human exposure.
The groups consist of a few elements. Is the low number of elements counterbalanced by a low variability within every group?
At line 292 the meaning of the sentence is too generic: not always the presence of methylation is strictly related to low expression: the inverse correlation between methylation and expression is not always a rule but a possibility which depends also on the methylation density within the promoter gene.
There is no information about the possibility to restore the physiological epigenetic landscape after suspension of the exposition.
A translational purpose for this kind of study would be very interesting. The epigenome of rats is not the same as humans. The authors could compare, for instance, the presence of CpG islands in the genes which are epigenetically modified in rats by hexanal respect to the same genes and/or pathways in humans.

Reviewer 2 ·

Basic reporting

In the current study, Cho et al has carried out methylome sequencing of hexanal treated rat lungs and made some relevant observations. The authors have demonstrated experience in these models and the studies follow similar pattern from their previous studies where they analyze mRNA levels. Overall, this study is of interest to the readers who study lung toxicity by volatile irritants.

The introduction Line no 40-58 are non-relevant and should be removed.
Grammatical errors and sentences without much information are found; Line 84-87.

Experimental design

The experimental set up is well thought out and the technologies used are recent and gives meaningful data.
Specific comments

In methods, mention reference/s for the protocol for methylated DNA extraction by fragmentation. Gel images for size determination should be shown as supplementary fig.
Line 207; add reference correctly.
Serum biochemical analysis data/ toxicological data is missing from the manuscript. Also, data not shown is not acceptable (lines 239,242). Add these data to supplementary info.

Validity of the findings

Fig 1 and Fig 2B are trivial. The readers would not gain much info from these figs.
There is no validation of the medip seq by bisulfite sequencing or methylation-specific PCR.
There is no validation of the genes correlated in medip vs mRNA array. qPCR should be carried out for at least some top hit genes.
Discussion Lines 317 to 336 is rather vague and do not convey anything and can be removed. There is no discussion about individual genes in the discussion.
Is there changes in corresponding levels of proteins after hexanal treatment? Discuss other studies.

Additional comments

The authors could rewrite the results and discussion to make the findings more clear to the readers. An illustration about what happens after hexanal exposure at gene level could be depicted based on the GO and other comparative analyses.

·

Basic reporting

Basic reporting
This paper is well-organized with no significant grammatical issues found. Sufficient background material is provided.

a) Minor Grammar Issues:
1) Line 123 should be changed to "in accordance with relevant guidelines"
2) Spelling error "diffierentiation" on line 309
3) Missing articles and Incorrect article usage throughout the paper. I suggest the authors re-run a basic grammar check.
4) line 373-4 "Using the DAVID functional annotation bioinformatics tool, we performed GO analysis was also performed"
5) Did the authors mean Benjamini-Hochberg on line 221?

b) Background context

1) A good background material is provided throughout out the introduction and discussion.
2). The authors have done a good job of organizing the paper professionally. I suggest the authors provide Tables in CSV or excel formats in the supplementary material.
3) Is it possible to label some/all of 73 genes in figure-2?

Experimental design

The article Fits well within the Aims and Scope of the journal.
The following improvements should be made to the manuscript :

1) Authors should clarify for which values of r and p, they call something as an anti-correlated gene.

2) On line 352-3 "we identified the differentially methylated genes of hexanal exposure showing a 3.0-fold-change (p < 0.05). "
Did the authors perform multiple test correction? I could find the relevant text of selection criteria in methods

Validity of the findings

The authors have done a good job of providing all underlying data needed to replicate the study. The tables and figures are sufficiently clear to understand the results

Additional comments

I think the authors have done a good job of presenting data and results for their claims. While I appreciate the overall effort, authors should following clarifications to the paper:
1) Criteria for an anti-correlated gene
2) Whether FDR adjustment was done in differentially methylated genes analysis.
I also suggest the following minor edits:
1) Fixing grammatical issues
2) Adding labels to figure-2
3) Clarity on criteria in the methods section

---

## Round 0.2 · accepted · Accept

You have adequately replied to the issues raised by the referees.

Reviewer 2 ·

Basic reporting

The authors have significantly improved the manuscript. No further comments.

Experimental design

No comment

Validity of the findings

No comment